# Serum Potassium Levels at Hospital Discharge and One-Year Mortality among Hospitalized Patients

**DOI:** 10.3390/medicina56050236

**Published:** 2020-05-14

**Authors:** Charat Thongprayoon, Wisit Cheungpasitporn, Sorkko Thirunavukkarasu, Tananchai Petnak, Api Chewcharat, Tarun Bathini, Saraschandra Vallabhajosyula, Michael A. Mao, Stephen B. Erickson

**Affiliations:** 1Division of Nephrology and Hypertension, Department of Medicine, Mayo Clinic, Rochester, MN 55905, USA; Thirunavukkarasu.Sorkko@mayo.edu (S.T.); chewcharat.api@mayo.edu (A.C); erickson.stephen@mayo.edu (S.B.E.); 2Division of Nephrology, Department of Internal Medicine, University of Mississippi Medical Center, Jackson, MS 39216, USA; wcheungpasitporn@gmail.com; 3Division of Pulmonary and Critical Care Medicine, Department of Medicine, Mayo Clinic, Rochester, MN 55905, USA; petnak@yahoo.com; 4Department of Internal Medicine, University of Arizona, Tuscon, AZ 85721, USA; tarunjacobb@gmail.com; 5Department of Cardiovascular Medicine, Mayo Clinic, Rochester, MN 55905, USA; Vallabhajosyula.Saraschandra@mayo.edu; 6Division of Nephrology and Hypertension, Mayo Clinic, Jacksonville, FL 32224, USA; mao.michael@mayo.edu

**Keywords:** hypokalemia, hyperkalemia, potassium, electrolytes, discharge, mortality

## Abstract

*Background and Objectives:* The optimal range of serum potassium at hospital discharge is unclear. The aim of this study was to assess the relationship between discharge serum potassium levels and one-year mortality in hospitalized patients. *Materials and Methods:* All adult hospital survivors between 2011 and 2013 at a tertiary referral hospital, who had available admission and discharge serum potassium data, were enrolled. End-stage kidney disease patients were excluded. Discharge serum potassium was defined as the last serum potassium level measured within 48 h prior to hospital discharge and categorized into ≤2.9, 3.0–3.4, 3.5–3.9, 4.0–4.4, 4.5–4.9, 5.0–5.4 and ≥5.5 mEq/L. A Cox proportional hazards analysis was performed to assess the independent association between discharge serum potassium and one-year mortality after hospital discharge, using the discharge potassium range of 4.0–4.4 mEq/L as the reference group. *Results:* Of 57,874 eligible patients, with a mean discharge serum potassium of 4.1 ± 0.4 mEq/L, the estimated one-year mortality rate after discharge was 13.2%. A U-shaped association was observed between discharge serum potassium and one-year mortality, with the nadir mortality in the discharge serum potassium range of 4.0–4.4 mEq/L. After adjusting for clinical characteristics, including admission serum potassium, both discharge serum potassium ≤3.9 mEq/L and ≥4.5 mEq/L were significantly associated with increased one-year mortality, compared with the discharge serum potassium of 4.0–4.4 mEq/L. Stratified analysis based on admission serum potassium showed similar results, except that there was no increased risk of one-year mortality when discharge serum potassium was ≤3.9 mEq/L in patients with an admission serum potassium of ≥5.0 mEq/L. *Conclusion:* The association between discharge serum potassium and one-year mortality after hospital discharge had a U-shaped distribution and was independent of admission serum potassium. Favorable survival outcomes occurred when discharge serum potassium was strictly within the range of 4.0–4.4 mEq/L.

## 1. Introduction

Potassium is a vital mineral that plays a versatile role in numerous cellular and enzymatic functions [1,2]. Plasma levels are normally maintained within a narrow range, measured at our institution from 3.5 to 5.2 mEq/L [2,3,4,5]. Fluctuations in serum potassium levels are minimized by an extremely organized and efficient homeostatic system—the primary orchestrator being the kidney [3]. The significance of abnormally elevated serum potassium levels is well known. Hyperkalemia can lead to muscle weakness, fatal arrhythmias, and even death [2]. Mild hyperkalemia or hypokalemia, however, can invoke less prominent symptoms and signs, and can even be subclinical [2,3,4,5].

Previous research, however, has shown that even mild potassium abnormalities have been associated with worse patient outcomes [4,5]. Recognizing the importance of potassium homeostasis, several investigators have strived to learn more about its clinical associations, and to decipher the optimal serum potassium range. Goyal et al. showed that in patients admitted for an acute myocardial infarction, there was an association between lower mortality in patients who maintained potassium levels between 3.5 and 4.5 mEq/L, compared to those who did not [6]. Multiple investigations of hospitalized patients have observed that higher or lower serum potassium levels at the time of hospital admission are associated with increased hospital mortality [4,5]. In addition, a meta-analysis of observational studies has shown an increase in arrhythmias and cardiovascular mortality with abnormal potassium levels [7].

Given that potassium abnormalities are associated with increased mortality, serum potassium is carefully monitored during hospitalization with a narrow potassium variability [4,5,8]. The appropriate correction of serum potassium during hospitalization has been shown to reduce mortality risk [9,10]. Nevertheless, many patients still have some degree of abnormal serum potassium at hospital discharge [9,10,11,12]. Thus, in this current study, we investigated the long-term effects of abnormal potassium levels at hospital discharge by assessing its association with one-year mortality.

## 2. Materials and Methods

### 2.1. Study Population

This is a single-center cohort study conducted at Mayo Clinic Hospital, Rochester, Minnesota, USA. All adult hospitalized patients who survived until hospital discharge from 2011–2013 were screened. Inclusion criteria consisted of patients who had at least two serum potassium measurements performed during their hospitalization: one potassium level was obtained within 24 h of hospital admission (representing admission serum potassium), and one potassium level was obtained within 48 h prior to hospital discharge (representing discharge serum potassium). End-stage kidney disease patients were excluded. Only the first hospital admission was analyzed for patients with multiple hospital admissions during the study period. This study was reviewed and approved by the Mayo Clinic institutional review board (IRB number 15-000024). The informed consent requirement was waived due to the minimal risk of the study, but all included patients provided research authorization for patient data use.

### 2.2. Data Collection

Discharge serum potassium was the main predictor of interest. The estimated glomerular filtration (eGFR) was calculated based on age, sex, race, and discharge serum creatinine, using the Chronic Kidney Disease Epidemiology Collaboration equation [13]. Principal diagnoses were categorized based on ICD-9 codes. The Charlson comorbidity score was computed to assess for the comorbidity burden of individual patients. In-hospital acute kidney injury was defined as an increase in serum creatinine of ≥0.3 mg/dL or 1.5 times from the baseline at any time during hospitalization [14]. The primary outcome was one-year mortality after hospital discharge. Patient vital statuses were obtained from our institutional registry and Social Security Death Index database.

### 2.3. Statistical Analysis

Continuous variables are presented as mean ± standard deviation (SD) and categorical variables as counts with percentages, respectively. Differences in clinical characteristics among discharge potassium groups were tested using ANOVA for continuous variables and a chi-square test for categorical variables. Discharge serum potassium was categorized into seven groups in order to assess the non-linear association with one-year mortality: ≤2.9, 3.0–3.4, 3.5–3.9, 4.0–4.4, 4.5–4.9, 5.0–5.4, and ≥5.5 mEq/L. A discharge serum potassium level of 4.0–4.4 mEq/L was selected as the reference group for outcome comparison because this range was associated with the nadir for one-year mortality. Patient survival was measured from hospital discharge and followed until death or one year after discharge. Patients who were lost to follow-up, or whose vital status was unknown, were censored at the date of their last inpatient/outpatient follow-up visit. One-year mortality risk was estimated using a Kaplan–Meier plot, and it was compared between discharge serum potassium groups using a log-rank test. A multivariable Cox proportional hazards analysis was constructed to analyze the independent risk of one-year mortality with discharge serum potassium, adjusting for age, sex, race, admission and discharge eGFR, principal diagnosis, Charlson comorbidity score, history of coronary artery disease, congestive heart failure, peripheral vascular disease, stroke, diabetes mellitus, chronic obstructive pulmonary disease, cirrhosis, in-hospital acute kidney injury, the use of renal replacement therapy, mechanical ventilation, vasopressor during hospitalization, and admission serum potassium. The interaction between admission and discharge serum potassium on mortality was also tested. Stratified analysis based on admission serum potassium (≤3.9, 4.0–4.9, and ≥5.0 mEq/L) was performed. A two-tailed p-value of less than 0.05 was considered statistically significant. All analyses were performed using JMP statistical software (version 10, SAS Institute, Cary, NC, USA, 2012).

## 3. Results

### 3.1. Clinical Characteristics

A total of 57,874 eligible patients were studied. The mean age was 63 ± 17 years and 54% were male. The mean admission serum potassium was 4.2 ± 0.6 mEq/L and the mean discharge serum potassium was 4.1 ± 0.4 mEq/L. A discharge serum potassium of ≤2.9, 3.0–3.4, 3.5–3.9, 4.0–4.4, 4.5–4.9, 5.0–5.4, and ≥5.5 mEq/L was seen in 0.1%, 5.0%, 29.0%, 42.0%, 20.0%, 4.0%, and 0.3% of patients, respectively. Clinical characteristics based on the discharge serum potassium levels are shown in Table 1.

### 3.2. Discharge Serum Potassium and One-Year Mortality

The estimated one-year mortality rate after discharge was 13.2%. There was a U-shaped association between either admission or discharge serum potassium and one-year mortality, with the nadir mortality associated with a discharge serum potassium of 4.0–4.4 mEq/L (Figure 1). 

The estimated one-year mortality was 29.8%, 16.9%, 13.3%, 11.4%, 14.2%, 19.7%, and 30.4% associated with a discharge serum potassium of ≤2.9, 3.0–3.4, 3.5–3.9, 4.0–4.4, 4.5–4.9, 5.0–5.4, and ≥5.5 mEq/L, respectively (Table 2 and Figure 2). 

When adjusted for potential confounders, including admission serum potassium, a discharge serum potassium below 4.0 mEq/L was progressively associated with an increased risk of one-year mortality, with hazard ratios of 1.15 (95% CI 1.08–1.23) for a discharge serum potassium of 3.5–3.9 mEq/L, 1.39 (95% CI 1.24–1.55) for 3.0–3.4 mEq/L, and 2.36 (95% CI 1.46–3.82) for ≤2.9 mEq/L, compared with the reference of 4.0–4.4 mEq/L. Similarly, a discharge serum potassium above 4.4 mEq/L was progressively associated with an increased risk of one-year mortality, with hazard ratios of 1.18 (95% CI 1.11–1.27) for a discharge serum potassium of 4.5–4.9 mEq/L, 1.50 (95% CI 1.34–1.67) for 5.0–5.4 mEq/L, and 2.22 (95% CI 1.65–2.98) for ≥5.5 mEq/L, compared with the reference of 4.0–4.4 mEq/L (Table 2).

### 3.3. Stratified Analysis Based on Admission Serum Potassium

In patients with an admission serum potassium of ≤3.9 or 4.0–4.9 mEq/L, a discharge serum potassium of ≤3.9 or ≥4.5 mEq/L were significantly associated with an increased risk of one-year mortality, compared with the reference of 4.0–4.4 mEq/L. In contrast, in patients with an admission serum potassium of ≥5.0 mEq/L, only a discharge serum potassium of ≥4.5 mEq/L was significantly associated with an increased risk of one-year mortality, compared with the reference of 4.0–4.4 mEq/L (Table 3). Statistical analysis showed a significant interaction between admission and discharge serum potassium on one-year mortality outcomes (*p* for interaction = 0.03).

## 4. Discussion

This cohort study demonstrates that hospital discharge serum potassium levels are associated with a U-shaped curve for one-year mortality, independent of admission serum potassium levels. Discharge serum potassium levels of 4.0–4.4 mEq/L were associated with the lowest one-year mortality, and there was a progressive increase in mortality with further deviation from this identified optimal serum potassium range. Discharge serum potassium levels of ≤2.9 and ≥5.5 mEq/L had the highest one-year mortality rates of 29.8% and 30.4%, respectively. Of note, even patients with discharge serum potassium levels typically described as normal, 3.5–3.9 mEq/L and 4.5–4.9 mEq/L, were associated with an increased risk of one-year mortality.

The impact of potassium on physiological function has been well recognized. It is also appreciated that serum potassium is only a surrogate marker for intracellular potassium, where the majority of potassium resides [15]. Potassium plays a central role in enzymatic functions. Even mild hypokalemia, when induced by thiazide diuretics, can lead to hyperglycemia through dysfunction of the potassium pump that regulates insulin secretion [16]. Studies have also shown that severe hyperkalemia or hypokalemia are causative factors in fatal arrhythmias due to alterations in the cardiac action potential, with the correction of serum potassium reducing the immediate risk of death [17,18,19]. Our study supports this finding, as even mild deviations in serum potassium from the optimal range were associated with increased mortality.

Although only approximately 9% of hospitalized patients had an abnormal serum potassium level at the time of hospital discharge, a significantly higher long-term mortality rate was observed in these patients. The associations between serum potassium level and clinical outcomes have been previously reported in several studies, specifically among patients with cardiovascular and renal diseases [4,7,20,21,22]. These previous studies demonstrated that both hypokalemia and hyperkalemia resulted in a higher mortality rate. Any deviation in serum potassium from a normal range, even if mild, should be taken seriously [23,24]. Our study also again raises the question of the optimal range for serum potassium, and we noted an increased risk of mortality with any deviation in discharge serum potassium from the reference range of 4.0–4.4 mEq/L, as graphically represented as the U-shaped mortality curve. The increased one-year mortality with even mild dyskalemia suggests that the “reference” serum potassium range may need reconsideration with a narrower optimal range than our current laboratory guidelines. A prior report in heart failure patients similarly demonstrated that discharge serum potassium had a U-shaped association with mortality risk [25].

Several studies have demonstrated a U-shaped association between admission serum potassium and mortality [4,20,22,26]. In contrast, this study investigated the impact of discharge serum potassium on long-term mortality among hospital survivors. While admission serum potassium is affected by acute illness and typically has not yet been altered by treatment, discharge serum potassium is often the result of medical treatment and a reflection of when medical providers have deemed patients safe for hospital discharge [25]. Therefore, the proportion of patients with abnormal serum potassium at hospital discharge was not unexpectedly smaller than those at hospital admission. Additionally, we found that a similar U-shaped association between discharge serum potassium and one-year mortality persisted even after adjustment for admission serum potassium. However, unlike admission serum potassium, where the optimal range associated with favorable survival was 4.0–4.9 mEq/L, the optimal discharge serum potassium level associated with favorable survival was in a narrower range of 4.0–4.4 mEq/L. Furthermore, we discovered that the magnitude of discharge serum potassium deviation from the optimal range had a greater impact on increased one-year mortality when compared with a similar degree of deviation in admission serum potassium.

Our study does have several limitations. First, it was performed at a large single referral center with a predominantly Caucasian population. Thus, generalizability to other patient populations is limited. Second, this study did not assess the association of the in-hospital potassium level trajectory and one-year mortality. This may impact the results of this study [27]. Third, minor serum potassium abnormalities may represent the disruption of potassium homeostasis and be an indirect indicator of subclinical organ dysfunction. As this was a retrospective study, there may have been unadjusted clinical variables that we did not account for. The data from this study were retrieved from an institutional electronic database. Unfortunately, some critical information, such as the causes of dyskalemia, diet, potassium supplement, and medications (e.g., beta-blockers, angiotensin converting enzyme inhibitors, angiotensin II receptor blockers, aldosterone antagonists, diuretics, non-steroidal anti-inflammatory drugs), were not available in our database and, therefore, we were not able to report them.

## 5. Conclusions

In conclusion, abnormal serum potassium levels at hospital discharge were associated with a U-shape mortality curve independent of admission potassium levels. Even mildly abnormal potassium levels at hospital discharge were associated with increased mortality. The highest long-term survival was found in patients who had a discharge serum potassium level of 4–4.4 mEq/L. 

## Figures and Tables

**Figure 1 medicina-56-00236-f001:**
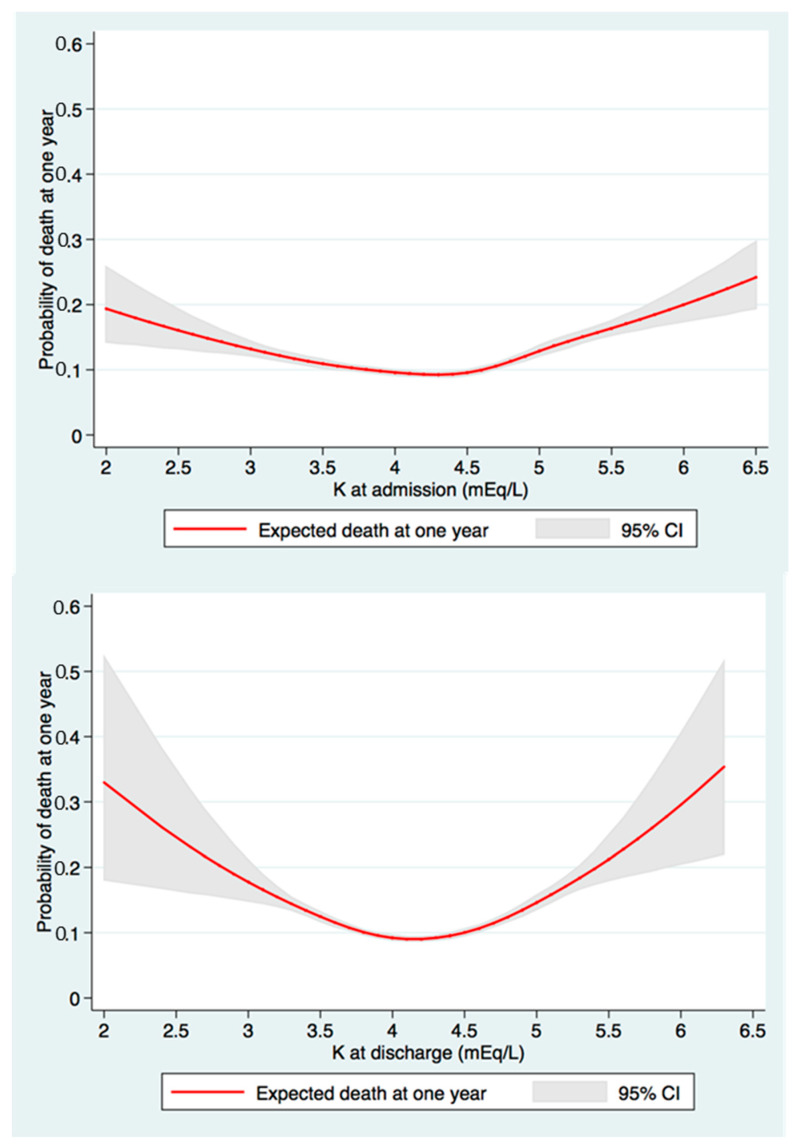
The restricted cubic spline demonstrated a U-shaped association between either admission (upper) or discharge (lower) serum potassium and one-year mortality.

**Figure 2 medicina-56-00236-f002:**
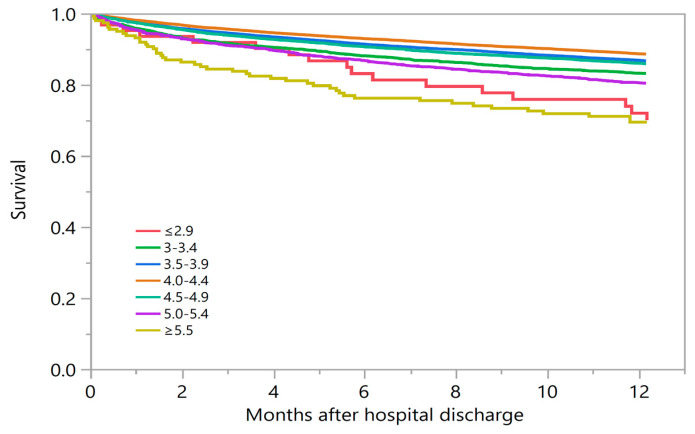
The Kaplan–Meier plot demonstrates one-year mortality based on discharge serum potassium levels.

**Table 1 medicina-56-00236-t001:** Baseline clinical characteristics.

Variables	All	Discharge Serum Potassium Level (mEq/L)							
		≤2.9	3.0–3.4	3.5–3.9	4.0–4.4	4.5–4.9	5.0–5.4	≥5.5	*p*
N	57,874	71	2666	16,789	24,493	11,370	2307	178	
Age (years)	63 ± 17	62 ± 17	63 ± 17	62 ± 18	63 ± 17	64 ± 17	65 ± 16	64 ± 16	< 0.001
Male	30,964 (54)	24 (34)	1113 (42)	8153 (49)	13,543 (55)	6647 (17)	1637 (59)	117 (66)	< 0.001
Caucasian	53,971 (93)	67 (94)	2454 (92)	15,549 (93)	22,897 (94)	10,663 (94)	2181 (95)	160 (90)	< 0.001
eGFR at admission (mL/min/1.73m^2^)	78 ± 26	82 ± 27	80 ± 28	81 ± 26	79 ± 26	74 ± 27	70 ± 28	62 ± 28	< 0.001
eGFR at discharge (mL/min/1.73m^2^)	81.3 ± 26.3	87.6 ± 27.1	85.6 ± 27.1	85.4 ± 25.7	81.5 ± 25.8	76.1 ± 26.4	70.1 ± 27.4	63.6 ± 28.3	< 0.001
Principal diagnosis - Cardiovascular - Endocrine/metabolic - Gastrointestinal - Hematology/oncology - Infectious disease - Respiratory - Injury/poisoning - Other	14,393 (25) 1547 (3) 5889 (10) 8685 (15) 1925 (3) 2548 (4) 9101 (16) 13,786 (24)	9 (13) 8 (11) 14 (20) 11 (16) 5 (7) 1 (1) 7 (10) 16 (23)	375 (14) 114 (4) 463 (17) 438 (16) 150 (6) 107 (4) 425 (16) 594 (22)	3446 (21) 458 (3) 2177 (13) 2516 (15) 707 (4) 699 (4) 2772 (17) 4014 (24)	6571 (27) 553 (2) 2228 (9) 3552 (15) 656 (3) 1011 (4) 3912 (16) 6010 (25)	3278 (29) 325 (3) 808 (7) 1769 (16) 343 (3) 557 (5) 1649 (15) 2641 (23)	675 (29) 79 (3) 181 (8) 369 (16) 63 (3) 154 (7) 314 (14) 472 (21)	39 (22) 10 (6) 18 (10) 30 (17) 1 (1) 19 (11) 22 (12) 39 (22)	< 0.001
Charlson score	1.8 ± 2.4	2.3 ± 2.7	2.0 ± 2.6	1.8 ± 2.4	1.8 ± 2.3	1.9 ± 2.4	2.3 ± 2.6	2.9 ± 2.8	< 0.001
Comorbidities - Coronary artery disease - Congestive heart failure - Peripheral vascular disease - Stroke - Diabetes mellitus - COPD - Cirrhosis	4565 (8) 4251 (7) 1836 (3) 4469 (8) 11,770 (20) 5334 (9) 1501 (3)	2 (3) 1 (1) 0 (0) 5 (7) 13 (18) 2 (3) 3 (4)	163 (6) 159 (6) 73 (3) 183 (7) 504 (19) 222 (8) 94 (4)	1097 (7) 1117 (7) 431 (3) 1187 (7) 3116 (19) 1370 (8) 440 (3)	1957 (8) 1768 (7) 772 (3) 1910 (8) 4830 (20) 2140 (9) 579 (2)	1060 (9) 958 (8) 434 (4) 961 (9) 2614 (23) 1268 (11) 286 (3)	267 (12) 231 (10) 113 (5) 203 (9) 636 (28) 299 (13) 90 (4)	19 (11) 17 (10) 13 (7) 20 (11) 57 (32) 33 (19) 9 (5)	< 0.001 < 0.001 < 0.001 < 0.001 < 0.001 < 0.001 < 0.001
Acute kidney injury in hospital	12,039 (21)	14 (20)	602 (23)	3075 (18)	4769 (20)	2746 (24)	751 (33)	82 (46)	< 0.001
Renal replacement therapy in hospital	243 (0.4)	0 (0)	8 (0.3)	58 (0.3)	97 (0.4)	57 (0.5)	20 (0.9)	3 (1.7)	0.001
Mechanical ventilation in hospital	9178 (16)	5 (7)	393 (15)	2625 (16)	3934 (16)	1796 (16)	397 (17)	28 (16)	0.08
Vasopressor use in hospital	4949 (9)	2 (3)	190 (7)	1353 (8)	2154 (9)	1005 (9)	227 (10)	18 (10)	0.001
Admission serum potassium (mEq/L)	4.2 ± 0.6	3.7 ± 0.7	3.8 ± 0.5	4.0 ± 0.5	4.2 ± 0.5	4.4 ± 0.6	4.6 ± 0.6	4.8 ± 0.7	< 0.001

Continuous data are presented as mean ± SD; categorical data are presented as count (%).

**Table 2 medicina-56-00236-t002:** Association between admission or discharge serum potassium levels and one-year mortality.

Serum Potassium Level (mEq/L)	One-Year Mortality (%)	Univariate Analysis		Multivariate Analysis	
		HR (95% CI)	*p*	Adjusted HR ^#^ (95 % CI)	*p*
**At admission**					
≤2.9	18.5%	1.67 (1.38–2.03)	<0.001	1.58 (1.30–1.92)	<0.001
3.0–3.4	14.8%	1.30 (1.17–1.45)	<0.001	1.20 (1.07–1.33)	0.001
3.5–3.9	13.1%	1.14 (1.07–1.22)	<0.001	1.14 (1.07–1.22)	<0.001
4.0–4.4	11.6%	1 (ref)	-	1 (ref)	-
4.5–4.9	13.1%	1.14 (1.07–1.22)	<0.001	1.04 (0.97–1.11)	0.28
5.0–5.4	17.3%	1.57 (1.42–1.72)	<0.001	1.19 (1.07–1.31)	0.001
≥5.5	22.6%	2.16 (1.90–2.47)	<0.001	1.43 (1.25–1.64)	<0.001
**At discharge**					
≤2.9	29.8%	2.78 (1.73–4.48)	<0.001	2.36 (1.46–3.82)	<0.001
3.0–3.4	16.9%	1.57 (1.41–1.75)	<0.001	1.39 (1.24–1.55)	<0.001
3.5–3.9	13.3%	1.19 (1.12–1.26)	<0.001	1.15 (1.08–1.23)	<0.001
4.0–4.4	11.4%	1 (ref)	-	1 (ref)	-
4.5–4.9	14.2%	1.27 (1.19–1.36)	<0.001	1.18 (1.11–1.27)	<0.001
5.0–5.4	19.7%	1.84 (1.65–2.06)	<0.001	1.50 (1.34–1.67)	<0.001
≥5.5	30.4%	3.10 (2.31–4.15)	<0.001	2.22 (1.65–2.98)	<0.001

# Adjusted for age, sex, race, admission and discharge eGFR, principal diagnosis, Charlson comorbidity score, coronary artery disease, congestive heart failure, peripheral vascular disease, stroke, diabetes mellitus, chronic obstructive pulmonary disease, cirrhosis, in-hospital acute kidney injury, the use of renal replacement therapy, mechanical ventilation, vasopressor during hospitalization, and admission or discharge serum potassium.

**Table 3 medicina-56-00236-t003:** Subgroup analysis based on admission serum potassium levels.

Serum Potassium Level at Discharge (mEq/L)	One-Year Mortality (%)	Univariate Analysis		Multivariate Analysis	
		HR (95% CI)	*p*	Adjusted HR^#^ (95 % CI)	*p*
**Admission serum potassium ≤ 3.9 mEq/L**					
≤3.9	14.6%	1.29 (1.17–1.42)	<0.001	1.23 (1.11–1.36)	<0.001
4.0–4.4	11.7%	1 (ref)	-	1 (ref)	-
≥4.5	15.4%	1.35 (1.18–1.54)	<0.001	1.27 (1.11–1.45)	<0.001
**Admission serum potassium 4.0–4.9 mEq/L**					
≤3.9	13.1%	1.23 (1.14–1.34)	<0.001	1.19 (1.10–1.29)	<0.001
4.0–4.4	10.8%	1 (ref)	-	1 (ref)	-
≥4.5	13.5%	1.28 (1.18–1.39)	<0.001	1.20 (1.11–1.31)	<0.001
**Admission serum potassium ≥ 5.0 mEq/L**					
≤3.9	16.6%	1.05 (0.83–1.31)	0.69	0.99 (0.79–1.25)	0.94
4.0–4.4	15.6%	1 (ref)	-	1 (ref)	-
≥4.5	21.8%	1.46 (1.25–1.71)	<0.001	1.44 (1.23–1.69)	<0.001

# Adjusted for age, sex, race, admission and discharge eGFR, principal diagnosis, Charlson comorbidity score, coronary artery disease, congestive heart failure, peripheral vascular disease, stroke, diabetes mellitus, chronic obstructive pulmonary disease, cirrhosis, in-hospital acute kidney injury, the use of renal replacement therapy, mechanical ventilation, and vasopressor during hospitalization.

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
