# Peer review of "Serum Potassium Levels at Hospital Discharge and One-Year Mortality among Hospitalized Patients"

_medicina, 2020, doi:10.3390/medicina56050236_

Round 1
Reviewer 1 Report
Dear Editors of Medicines
I have carefully reviewed the manuscript entitled “Serum Potassium Levels at Hospital Discharge and One-Year Mortality among Hospitalized Patients” (medicina-785783) by Charat Thongprayoon.
This single-center retrospective study enrolled 57,874 adult hospitalized patients (mean age, 63±17 years, male 54%) who survived until hospital discharge from years 2011-2013 and had two serum potassium measurements at both time points of admission and discharge.
The authors categorized patients into seven groups by “discharge serum potassium” (≤2.9, 3.0–3.4, 3.5–3.9, 4.0–4.4, 4.5–4.9, 5.0–5.4, and ≥5.5 mEq/L), an disclosed a U-shaped association between discharge serum potassium and one-year mortality, with the nadir mortality in the discharge serum potassium of 4.0-4.4 mEq/L.
By using the multivariable Cox proportional hazard analysis, the authors found that both discharge serum potassium ≤3.9 mEq/L and ≥4.5 mEq/L were significantly associated with increased one-year mortality, compared with the discharge serum potassium of 4.0-4.4 mEq/L. The results persisted in the stratified analysis based on admission serum potassium.
Generally speaking, this is a well-written manuscript.
The topic of the current study is interesting and of clinical relevance.
The background of the study, study design, data presentation and interpretation, as well as English writing are all good.
However, one minor concern needs to be addressed.
#1. What is the 1-year mortality in the current study? I am confused by the two different data (13.2%. in “Abstract, line 31, pp.1,” versus 10.5% in “3.2. Discharge Serum Potassium and One-Year Mortality, line 122, pp.5.”)
Author Response
Response to Reviewer#1
Comment
I have carefully reviewed the manuscript entitled “Serum Potassium Levels at Hospital Discharge and One-Year Mortality among Hospitalized Patients” (medicina-785783) by Charat Thongprayoon.
This single-center retrospective study enrolled 57,874 adult hospitalized patients (mean age, 63±17 years, male 54%) who survived until hospital discharge from years 2011-2013 and had two serum potassium measurements at both time points of admission and discharge.
The authors categorized patients into seven groups by “discharge serum potassium” (≤2.9, 3.0–3.4, 3.5–3.9, 4.0–4.4, 4.5–4.9, 5.0–5.4, and ≥5.5 mEq/L), an disclosed a U-shaped association between discharge serum potassium and one-year mortality, with the nadir mortality in the discharge serum potassium of 4.0-4.4 mEq/L.
By using the multivariable Cox proportional hazard analysis, the authors found that both discharge serum potassium ≤3.9 mEq/L and ≥4.5 mEq/L were significantly associated with increased one-year mortality, compared with the discharge serum potassium of 4.0-4.4 mEq/L. The results persisted in the stratified analysis based on admission serum potassium.
Generally speaking, this is a well-written manuscript.
The topic of the current study is interesting and of clinical relevance.
The background of the study, study design, data presentation and interpretation, as well as English writing are all good.
However, one minor concern needs to be addressed.
Response: We thank you for reviewing our manuscript and for your critical evaluation.
Comment #1
What is the 1-year mortality in the current study? I am confused by the two different data (13.2%. in “Abstract, line 31, pp.1,” versus 10.5% in “3.2. Discharge Serum Potassium and One-Year Mortality, line 122, pp.5.”)
Response: We appreciate the reviewer’s important comment. The difference in reported percentage of 1-year mortality in abstract and result section was due to the different determination methods. One-year mortality was estimated to be 13.2% using Kaplan-Meier method, censoring patient who lost follow-up before 1 year to account for varied follow-up time of each patient. In contrast, 10.5% was calculated based on the percentage of patients who died in one year but this number did not account for incomplete follow-up of some patients. To avoid the confusion, we revised statements in result section to make them consistent with abstract section.
We greatly appreciated the editor and reviewers’ time and suggestions to improve our manuscript.
Reviewer 2 Report
The authors examined the relationship between discharge serum potassium levels and 1-year survival from a single institute. They found a U-shape relationship with the nadir level at 4.0 – 4.4 meq/L. Overall, the idea is not without interest. I have the following suggestions.
- The authors explicitly stated that they focused only on the admission and discharge serum potassium levels, which is a clever approach. However, this seems to overlook the importance of serum potassium changes during the hospitalization period, which may have more pathophysiological meanings and values. Why didn’t the authors choose potassium levels at other time points, use potassium level variations, or even the potassium trajectory during the hospitalization period? This should be justified very clearly and reasonably in their introduction. Alternatively, the suggestions above can be adopted with more results provided.
- The definitions of several clinical variables need clarification. For acute kidney injury, when did that happen during the hospitalization period? What were the baseline creatinine levels of these patients? Could baseline eGFR be provided in addition to discharge eGFR and factored into all the regression analyses done? In addition, there seem to be missing variables which are needed for adjustment as well. For example, medication use that tend to affect serum potassium levels and outcomes at the same time, such as beta-blockers, ACEi, ARB, spironolactone, different types of diuretics, NSAIDs, etc. (this is very important, must be added). Also, what about treatment course variables during the hospitalization, such as vasopressor use, acute dialysis or not? What about other vital laboratory profiles that frequently vary simultaneously with serum potassium while influence outcome, such as albumin (important, must be considered)?
- Please specify which variables were included in different Cox models (there were multiple models in this study); the current manuscript only described “pre-specified variables in Table 1” in the method section, which is too vague and should be corrected.
- Please amend the word “GFR” throughout the manuscript text and tables; it is “eGFR” being used, not GFR which requires measurement.
- Please provide a restricted cubic spline plot using admission potassium levels as well (like Figure 1) for comparison.
Author Response
Response to Reviewer#2
Comment
The authors examined the relationship between discharge serum potassium levels and 1-year survival from a single institute. They found a U-shape relationship with the nadir level at 4.0 – 4.4 meq/L. Overall, the idea is not without interest. I have the following suggestions.
Response: We thank you for reviewing our manuscript and for your critical evaluation.
Comment #1
The authors explicitly stated that they focused only on the admission and discharge serum potassium levels, which is a clever approach. However, this seems to overlook the importance of serum potassium changes during the hospitalization period, which may have more pathophysiological meanings and values. Why didn’t the authors choose potassium levels at other time points, use potassium level variations, or even the potassium trajectory during the hospitalization period? This should be justified very clearly and reasonably in their introduction. Alternatively, the suggestions above can be adopted with more results provided.
Response: We appreciate the reviewer’s important comment. We agree with the reviewer and thus have additionally revised our introduction to emphasize an important on the impact of serum potassium level at hospital discharge and one-year mortality. Trajectory of potassium during the hospitalization is tightly controlled/corrected with a minimal potassium variability by physician’s treatment, and affected by medication/potassium replacement. Data on medication were limited in our database and thus we also added this point in our limitation of the study as well.
The following text has been added in the introduction as suggested.
“Given potassium abnormalities are associated with increased mortality, serum potassium is carefully monitored during hospitalization with a narrow potassium variability (4, 5, 8). Appropriate correction of serum potassium during hospitalization has been shown to reduce mortality risk (9, 10). Nevertheless, many patients still have some degrees of abnormal serum sodium at hospital discharge (9-12). Thus, in this current study, we investigated the long-term effects of abnormal potassium levels at hospital discharge by assessing its association with one-year mortality.”
Comment #2
The definitions of several clinical variables need clarification. For acute kidney injury, when did that happen during the hospitalization period? What were the baseline creatinine levels of these patients? Could baseline eGFR be provided in addition to discharge eGFR and factored into all the regression analyses done? In addition, there seem to be missing variables which are needed for adjustment as well. For example, medication use that tend to affect serum potassium levels and outcomes at the same time, such as beta-blockers, ACEi, ARB, spironolactone, different types of diuretics, NSAIDs, etc. (this is very important, must be added). Also, what about treatment course variables during the hospitalization, such as vasopressor use, acute dialysis or not? What about other vital laboratory profiles that frequently vary simultaneously with serum potassium while influence outcome, such as albumin (important, must be considered)?
Response: In-hospital acute kidney injury was defined as an increase in serum creatinine of ≥0.3 mg/dL or 1.5 times from baseline at any time during hospitalization.
The data on baseline eGFR at hospital admission, the use of renal replacement therapy, and vasopressor was added in Table 1, and was included in multivariable analysis, as suggested.
The data on medication was not available in our database and we were not able to report. The following statements have been added to the limitation.
“The data from this study were retrieved from institutional electronic database. Unfortunately, some critical information, such as the causes of dyskalemia, diet, potassium supplement, and medications (e.g. beta-blockers, angiotensin converting enzyme inhibitors, angiotensin II receptor blockers, aldosterone antagonists, diuretics, non-steroidal anti-inflammatory drug), were not available in our database and, therefore, we were not able to report them.”
Only 19% (n=11160) of this cohort had available serum albumin measurement during hospitalization. As majority of patients had missing serum albumin value, we did not include serum albumin in the analysis.
Comment #3
Please specify which variables were included in different Cox models (there were multiple models in this study); the current manuscript only described “pre-specified variables in Table 1” in the method section, which is too vague and should be corrected.
Response: The following statements have been revised to describe adjusting variables in the multivariable Cox model.
Cox proportional hazard analysis was constructed to analyze the independent risk of one-year mortality with discharge serum potassium, adjusting for age, sex, race, admission and discharge eGFR, principal diagnosis, Charlson comorbidity score, history of coronary artery disease, congestive heart failure, peripheral vascular disease, stroke, diabetes mellitus, chronic obstructive pulmonary disease, cirrhosis, in-hospital acute kidney injury, the use of renal replacement therapy, mechanical ventilation, vasopressor during hospitalization, and admission serum potassium.
Comment #4
Please amend the word “GFR” throughout the manuscript text and tables; it is “eGFR” being used, not GFR which requires measurement.
Response: This has been changed as suggested.
Comment #5
Please provide a restricted cubic spline plot using admission potassium levels as well (like Figure 1) for comparison.
Response: We agree with the reviewer. The restricted cubic spline of the association between admission serum potassium and one-year mortality was added in the manuscript as suggested.
(please kindly find added figure in PDF as suggested)
We greatly appreciated the editor and reviewers’ time and suggestions to improve our manuscript.
Round 2
Reviewer 2 Report
Thank you. The authors have made satisfactory changes to their manuscript.